# The Neuro-Ischaemic Charcot Foot: Prevalence, Characteristics and Severity of Peripheral Arterial Disease in Acute Charcot Neuro-Arthropathy

**DOI:** 10.3390/jcm11216230

**Published:** 2022-10-22

**Authors:** Marco Meloni, Raju Ahluwalia, Alfonso Bellia, Enrico Brocco, Michela Di Venanzio, Aikaterini Andreadi, Laura Giurato, Valeria Ruotolo, Nicola Di Daniele, Davide Lauro, Luigi Uccioli

**Affiliations:** 1Department of Systems Medicine, University of Rome “Tor Vergata”, 00133 Rome, Italy; 2Division of Endocrinology and Diabetology, Department of Medical Sciences, Fondazione Policlinico “Tor Vergata”, 00133 Rome, Italy; 3Department of Orthopaedics, King’s College Hospital, London & King’s College London & Royal College of Surgeons of England, London WC2A 3PE, UK; 4Diabetic Foot Centre, Abano Terme Polyclinic, 35031 Abano Terme, Italy; 5UOSD Diabetologia San Camillo De Lellis Hospital, 02100 Rieti, Italy; 6Division of Internal Medicine-Hypertension, Department of Medical Sciences, Fondazione Policlinico “Tor Vergata”, 00133 Rome, Italy; 7CTO Andrea Alesini Hospital, Division of Endocrinology and Diabetes, Department of Systems Medicine, University of Rome Tor Vergata, 00145 Rome, Italy

**Keywords:** diabetes, diabetic foot, Charcot neuro-arthropathy, peripheral arterial disease

## Abstract

The study aimed to evaluate the prevalence, characteristics and outcomes of patients affected by Charcot neuro-arthropathy (CN) and peripheral arterial disease (PAD) compared to CN without PAD. Consecutive patients presenting with an acute CN were included. The sample size was calculated by the power analysis by adopting the two-tailed tests of the null hypothesis with alfa = 0.05 and a value of beta = 0.10 as the second type error and, therefore, a test power equal to 90%. Seventy-six patients were identified. Twenty-four patients (31.6%) had neuro-ischaemic CN; they were older (66 vs. 57yrs), *p* = 0.03, had a longer diabetes duration (19 vs. 14yrs), *p* < 0.001, and more cases of end-stage-renal-disease (12.5 vs. 0%), *p* = 0.04 and ischaemic heart disease (58.3 vs. 15.4%), *p* < 0.0001 than neuropathic CN. Fifty patients (65.8%) had concomitant foot ulcers, 62.5% and 67.3% (*p* = 0.3), respectively, in CN with and without PAD. Neuro-ischaemic CN show arterial lesions of 2.9 vessels, and PAD was located predominantly below-the-knee (75%) but not below-the-ankle (16.7%). The outcomes for neuro-ischaemic and neuropathic CN patients were, respectively: wound healing (86.7 vs. 94.3%), *p* = 0.08; minor amputation (25 vs. 7.7%), *p* = 0.003; major amputation (8.3 vs. 1.9%), *p* = 0.001; hospitalization (75 vs. 23%), *p* = 0.0001. The study showed a frequent association between CN and PAD, leading to a neuro-ischaemic Charcot foot type. Neuro-ischaemic CN leaded to an increased risk of minor and major amputation and hospitalization, compared to neuropathic CN.

## 1. Introduction

Charcot neuro-arthropathy (CN) is a complex and severe complication of diabetic neuropathy associated with an increased risk of morbidity and mortality [1,2]. The annual incidence of diabetic foot ulcers (DFUs) in patients with CN is estimated around 17%, and the risk of developing a new active DFU in the case of bony deformities is 4 times higher if compared to diabetic patients without CN [3,4]. The prevalence of peripheral arterial disease (PAD) in patients with diabetes and foot problems is reported to be >50% [5,6], and PAD is independent predictor of negative outcomes including non-healing, amputation and mortality [7,8]. The form of atherosclerotic disease in diabetes usually involves infra-popliteal vessels in comparison to those without diabetes [9] and recently a more frequent involvement of foot arteries has been documented, especially in the case of concomitant renal failure requiring dialysis [10,11].

Until recently, CN has been thought of as a purely neuropathic syndrome and PAD was considered protective against the onset of CN [12]. Accordingly, PAD in CN was commonly less investigated, although more recently some authors have reported that PAD may be a concomitant condition in CN. Today, only a few authors have investigated the presence of PAD in CN patients, identifying the prevalence range of PAD from 5 to 40% [13,14]. These differences may reflect the non-homogeneous inclusion criteria and different diagnostic methods used to detect PAD.

A few studies have analysed the relationship between PAD in CN, specifically describing the distribution and severity, and the prognosis of neuro-ischaemic DFUs in patients presenting with CN. The aim of this study was to evaluate the prevalence, characteristics and the outcomes of patients affected by CN and PAD referring to a specialized third level diabetic foot service (DFS).

## 2. Materials and Methods

### 2.1. Patient Selection

This study is a retrospective study conducted in a single tertiary level DFS, assessing consecutive patients from January 2019 to February 2022, diagnosed with diabetes and acute CN stage 0 and 1 were included.

Patients excluded were patients without diabetes, those presenting with an extensive tissue loss and infection or those with fractures or dislocations of the osseous structures not suitable for limb salvage, and those with chronic CN, as reported in Table 1.

### 2.2. Diagnosis of CN

The diagnosis of CN was performed through clinical and radiological findings. Clinical findings included marked swollen, warm, and sometimes redness associated or not with mild pain or discomfort in patients with a documented peripheral sensory neuropathy [15,16]. Clinical evaluation was completed by radiological investigation performed by X-ray and magnetic resonance imaging (MRI) [15]. Radiological abnormalities for CN were defined by the presence of bone oedema, acute fragmentations and dislocations [15,17].

### 2.3. Evaluation of Peripheral Arterial Disease (PAD)

PAD was defined as either no palpable distal pedal pulses, TcPO2 < 50 mmHg [18,19] and arterial stenosis/occlusions documented by ultra-sound duplex. In the case of critical limb ischaemia (CLI) defined TcPO2 values < 30 mmHg, the presence of stenosis/occlusions documented by ultra-sound duplex and active DFU, lower limb revascularization was indicated.

### 2.4. Patient Treatment

At the time of admission, demographic and clinical data as well as DFU characteristics were recorded. All patients followed standard institutional assessment and management protocols as summarized here:

### 2.5. DFU Management

In the case of concomitant active DFU, all patients were managed by a pre-set limb salvage protocol following International Working Group on the Diabetic Foot (IWGDF) Guidelines [18], including the restoration of foot perfusion in the case of peripheral ischaemia, antibiotic therapy (and surgery if required) in the case of infection, off-loading of an affected limb, management of diabetes and comorbidities, ulcer debridement and local wound care based on the best evidence recommendations [18]. Revascularization procedures were performed in respect to foot condition, vessels affected and patient’s general condition by either endovascular or surgical (by-pass) procedure [18,20].

### 2.6. Off-Loading Protocol

All patients received off-loading both for treating CN and for relieving pressure and trauma in the ulcer area according to the ulcer location, the presence of ischaemia or infection (isolated or in conjunction) and individual needs [18]. Patients without active DFU and without PAD were treated by total contact cast (TCC) application; patients with active non-infected DFU and without PAD were treated by TCC or a removable high-knee cast (RKC) according to clinical evaluation; patients with infected DFU with or without PAD were treated by RKC; patient with PAD with or without active DFU were treated with RKC [18]. In each condition, a total relief of limb pressure was recommended until the resolution of acute CN [15].

### 2.7. Assessment of Outcomes of Interest

The primary aim was to define the prevalence of PAD. Accordingly, patients were divided in two groups, those with PAD (neuro-ischaemic CN) and those without (neuropathic CN). Healing, minor amputation, major amputation, and hospitalization were reported and compared between groups. Ulcer healing was taken to be complete epithelialization of the target wound, and maintenance of the closed healed epithelized surface for a minimum of 2 weeks. Minor amputation was considered any amputation below-the-ankle (digital, ray, metatarsal, Lisfranc, Chopart), and major amputation was considered any amputation above the ankle.

In patients with PAD, the characteristics and severity of PAD were described. The vessels affected, the localization of stenosis and/or occlusions, and the rate of revascularization were reported. All patients were followed up for 24 weeks.

### 2.8. Statistical Analysis

Statistical analysis was performed by SAS (JMP12; SAS Institute, Cary, NC, USA) for the personal computer. The sample size was calculated by the power analysis by adopting the two-tailed test of the null hypothesis with alfa = 0.05 and a value of beta = 0.10 as the second type error and, therefore, a test power equal to 90%. Continuous variables were expressed as the mean ± SEM. The Shapiro-Wilk test was used to statistically assess the normal distribution of the data. Comparisons between continuous variables were performed using the independent Student *t*-test. Categorical data were analysed using the chi square test. *p* < 0.05 was considered statistically significant.

## 3. Results

### 3.1. Demographics and Patient Characteristics at the Inclusion Time

In total, 1184 of new patients were seen with a new acute diabetic foot episode during the study period. Among those, 76 patients were included in the study, leaving an incidence of 6.4% (n = 76/1184) patients with a new episode of acute CN. The mean age was 60 ± 14 years; 44 (57.9%) were male, 66 (86.8%) were affected by Type 2 Diabetes with a mean duration of 16 ± 6 years (see Table 2). In total, 24 patients (31.6%) had a diagnosis of PAD and were considered neuro-ischaemic CN patients; they were observed to be older (66 vs. 57) (*p* = 0.03), had a longer diabetes duration (19yrs vs. 14yrs) (*p* < 0.001), and more established cases of end-stage-renal-disease (ESRD) (12.5 vs. 0%) (*p* = 0.04) and ischaemic heart disease (IHD) (58.3 vs. 15.4%) (*p* < 0.0001) than those considered to be neuropathic CN patients.

Among the whole population, there was a prevalence of patients with CN type 1 in comparison to type 0 (72.4 vs. 27.6%). In addition, type 1 CN showed more cases of PAD than type 0 (32.7 vs. 28.6%).

Fifty patients (60.8%) had concomitant foot ulcer at the time of presentation (see Table 3). The rate of concomitant foot ulcers was less in patients with neuro-ischaemic CN in comparison to neuropathic CN (62.5% vs. 67.3%) (*p* = 0.3) (see Table 3). The ulcers were studied carefully and found to be related to pressure or trauma due to not adequate shoes, and they were not related to the onset of concomitant CN.

### 3.2. Assessment of Outcomes of Interest

Overall, 46 (92%) patients with DFU healed, 10/76 (13.1%) had minor amputation, 3/76 (3.9%) had major amputation and 28/76 (36.8%) required hospitalization. Finite analysis showed the outcomes for neuro-ischaemic and neuropathic CN patients were, respectively, different: DFU healing 13/15 (86.7%) vs. 33/35 (94.3%), *p* = 0.08; minor amputation 6/24 (25%) vs. 4/52 (7.7%), *p* = 0.003; major amputation 2/24 (8.3%) vs. 1/52 (1.9%), *p* = 0.001; hospitalization 18/24 (75%) vs. 12/52 (23%) *p* = 0.0001 (see Table 4). The revascularisation rate in the neuro-ischaemic CN patients was 14/24 (58.3%).

Three cases of major amputation were related to the progression of infection in patients with acute CN combined with a DFU at presentation with deep tissue and bone involvement which did not allow for limb salvage despite standardized treatment.

### 3.3. Anatomical Characteristics and Distribution of PAD in Those Presenting with Acute CN

Patients with neuro-ischaemic CN had a mean TcPO2 of 33+/−5 mmhg. They were observed to have arterial tree stenosis/occlusions of 2.9 (min 1–max 6) vessels in the CN affected leg. These changes were considered to be diagnostic of PAD, and 19/24 (79.2%) reported the involvement of femoral-popliteal axis, 18/24 (75%) below-the-knee (BTK) arteries, and 4/24 (16.7%) below-the-ankle (BTA) arteries. We observed that 14/24 (58.3%) patients underwent lower limb revascularization as part of their treatment. See Table 5.

## 4. Discussion

Epidemiological data on the presence, characteristics and impact of PAD in CN patients is very rare and under reported. This study aimed to evaluate the prevalence of PAD in patients with CN and its impact in this specific group of subjects. The first data is that in our cohort, where the rate of neuro-ischaemic CN was approximately 31%. The reported data is similar to two previous studies in which the prevalence of PAD in CN patients using ABI was, respectively, of 27% and 25% [2,3]. The data presented here is higher than that reported in a previous Italian study [21] in which the prevalence was 11%, but only hospitalized patients were considered. Even so, Wukich et al. observed the prevalence of PAD could be as high as 40% in this group [13], whilst Oriali et al. observed it reaching 66% [14]. These differences may represent the heterogeneity in inclusion criteria of samples and methods used to investigate PAD among included patients.

In the current study, neuro-ischaemic CN patients were significantly older and observed to have a greater mean duration of diabetes. They presented a greater number of concomitant co-morbidities such as IHD and ESRD in comparison to neuropathic CN patients. These different features between neuro-ischaemic and neuropathic CN patients are similar to those reported in previous studies in which ischaemic and neuropathic DFU subjects without CN were compared [5,7].

The characteristics of PAD in this cohort of CN patients show a milder form of ischaemic disease at presentation, however, caution is required as the hyperdynamic in acute CN flow may alter normal flow patterns. Even so, there are now a number of studies identifying PAD in CN, and some suggest a less severe pattern of PAD in comparison to the usual characteristics of PAD in neuro-ischaemic DFUs patients without CN. We observed that neuro-ischaemic CN patients had a mean of 3 affected vessels, 75% had BTK arterial disease and approximately only 17% had BTA arterial disease. This is in contrast with recent studies examining DFU and PAD without CN, in which a mean of 4 vessels were affected, and the rate of BTK and BTA arterial disease was, respectively, approximately >90% and 40% [10,11,22].

A less severe phenotype of PAD in CN patients in comparison to those patients with diabetes without CN was also reported by Çildag et al. [23]. In their study, they evaluated patient angiograms, finding 12.8% of cases with only stenosis and 75.6% with stenosis and occlusions. Based on the distribution of infra-popliteal artery, 30.8% of patients reported only one occluded artery, 37.2% of patients two occluded arteries, and 10.3% of patients three occluded arteries. In contrast, patients without CN are reported to have 50% occurrence of two occluded arteries and in 14.6% of cases three occluded arteries [23]. They found the anterior tibial artery resulted the most involved artery (23.1%) [23].

As a consequence, neuro-ischaemic CN patients, even with a milder pattern of PAD, are observed to have higher rates of minor amputation (25 vs. 7.7%, *p* = 0.003), major amputation (8.3 vs 1.9%, *p* = 0.001) and higher need of hospitalization (75 vs. 23%, *p* = 0.0001) in comparison to neuropathic CN. Whilst the higher rate of hospitalization may be related to the need for revascularization in the sub-group of patients with PAD and CLI (58.3%). The rate of DFU was more less similar in patients with and without PAD (approximately 62% and 67%, respectively), and although neuropathic CN reported higher rate of DFU healing, there was not a statistically significant difference in the healing rate between the groups (86.7 vs. 94.3%, *p* = 0.08). 

The higher rate of minor and major amputation reinforces the concept that PAD increases the risk of amputation, particularly in the case of concomitant DFU, in comparison to the presence of pure peripheral neuropathy where healing potential is greater [5,7]. The rate of major amputation in our cohort of patients is similar to that reported by Orioli et al. (8 vs. 10%) [14] but lower than data reported by Cates et al. (8 vs. 37%) [24], although in that last mentioned study they included only patients who underwent Charcot reconstruction, therefore with an additional risk for a negative outcome [24].

Although the concomitant condition of Charcot neuropathy per se increases the risk of amputation [1,2], the rate of major amputation in neuro-ischaemic CN patients was observed to be lower than usual data reported in the literature on patients with diabetic foot problems and PAD without CN (8 vs. 7–30%) [10,23,25,26,27,28,29], especially if compared to patients with “no-option” CLI [22]. We would add the 3 patients undergoing major amputation in neuro-ischaemic CN patients were related to the development and progression of infection and not to untreatable lower limb ischaemia. Even so, PAD does have a negative role as it increases the risk of minor amputations, therefore these patients may need in depth evaluation and close monitoring during standard CN treatment.

In summary, we confirm a frequent association between CN and PAD. Despite several differences between the methods used for diagnosing PAD, in the current and previous studies, the association of PAD and CN is neither uncommon, and nor considered a protective factor for the onset of CN. The presence of PAD may increase the risk of major amputation if compared to CN without PAD. Nonetheless, the pattern of PAD in CN patients shows less penetrance compared to patients without CN both in terms of arterial disease distribution, the need for revascularization, amputation, and healing of DFU.

Therefore, the authors retain that PAD may develop in patients with diabetes and CN such as in those with diabetes without CN usually in relation to the presence of concomitant cardiovascular risk factors such as hypertension, dyslipidemia, and renal failure. Finally, the decreased severity of PAD we reported in our sample, specifically documented by the few cases of BTA arterial disease, may be related to the very low rate of ESRD patients. The involvement of BTA arteries in patients with diabetes is usually related to the concomitant presence of ESRD, as reported in previous study [10,11,30].

The study is a monocentric and retrospective study. In addition, the sample included only patients with CN and there is not a comparison of PAD between those with or without CN, but only a comparison with data extrapolated from the scientific literature. A larger multi-centre population may be useful to reinforce these data, but the low incidence of CN may limit this and as the current observations are in keeping with the literature can be considered acceptable. We appreciate a large proportion of patients presented with an ulcer, but we believe this may represent the patients first recognition of the foot abnormality as an ulcer but not the underlying CN process.

## 5. Conclusions

This study describes the prevalence, characteristics and outcomes of patients with CN and PAD, namely a “neuro-ischaemic CN”. Therefore, patients presenting with CN, should be investigated for the presence of PAD and managed accordingly. Understanding the impact of PAD in CN patients, both in terms of prevalence and severity, will help risk stratification, improve management strategies and outcomes and better define future perspectives in long-term risk management.

## Figures and Tables

**Table 1 jcm-11-06230-t001:** Definition of inclusion and exclusion criteria.

Inclusion Criteria
Diabetes
Acute CN (stage 0 or 1)
Presence or absence of DFU
**Exclusion Criteria**
Absence of diabetes
Chronic CN
Extensive foot tissue or foot infection not suitable for limb salvage
Foot/ankle fractures or dislocations of the osseous structures not suitable for limb salvage

CN: Charcot neuro-arthropathy. DFU: diabetic foot ulcer.

**Table 2 jcm-11-06230-t002:** Baseline demographic and clinical characteristics.

Variable	Whole Population Included (N = 76)	PAD Yes (N = 24)	PAD No (N = 52)	*p* Value
Sex (male)	44 (57.9%)	16 (72.9%)	28 (53.8%)	0.001
Age (years)	60 ± 14	66 ± 18	57 ± 13	0.03
Diabetes type (2)	66 (86.8)	18 (75%)	48 (92.3)	0.002
Diabetes duration (years)	16 ± 6	19 ± 4	14 ± 7	0.001
HbA1c (mmol/mol)	64 ± 28	62 ± 24	65 ± 30	0.7
ESRD	3 (4%)	3 (12.5%)	0 (0%)	0.04
IHD	22 (30.5%)	14 (58.3%)	8 (15.4%)	0.0001
Hypertension	73 (96%)	22 (91.7%)	51/52 (98%)	0.6
Dyslipidemia	51 (67.1%)	15(62.5%)	36 (69.2%)	0.8
Active Smoke	8 (10.5%)	3 (12.5%)	5 (9.6%)	0.9
Previous amputation	4 (5.3%)	3 (12.5%)	1 (1.9%)	0.06
CN type 0	21 (27.6%)	6 (28.6%)	15 (71.4%)	0.0001
CN type 1	55 (72.4%)	18 (32.7%)	37 (67.3%)	0.0006

PAD: peripheral arterial disease. ESRD: end-stage-renal-disease. IHD: Ischaemic heart disease. In the comparison between groups (CN with PAD vs CN without PAD), *p* < 0.05 was considered statistically significant.

**Table 3 jcm-11-06230-t003:** Association of ulcer and CN.

Variable	Whole Population Included (N = 76)	PAD Yes (N = 24)	PAD No (N = 52)	*p* Value
DFU (yes)	50/76 (65.8%)	15/24 (62.5%)	35/52 (67.3%)	0.3
DFU Duration (weeks)	7 ± 5	6 ± 4	8 ± 5	0.6

PAD: peripheral arterial disease. *p* < 0.05 was considered statistically significant.

**Table 4 jcm-11-06230-t004:** Outcomes of interest at 24 weeks.

Variable	Whole Population Included (N = 76)	PAD Yes (N = 24)	PAD No (N = 52)	*p* Value
DFU healing	46/50 (92%)	13/15 (86.7%)	33/35 (94.3%)	0.08
Minor amputation	10/76 (13.1%)	6/24 (25%)	4/52 (7.7%)	0.003
Major amputation	3/76 (3.9%)	2/24 (8.3%)	1/52 (1.9%)	0.001
Hospitalization	28/76 (36.8%)	18/24 (75%)	12/52 (23%)	0.0001

PAD: peripheral arterial disease. In the comparison between groups (CN with PAD vs CN without PAD) *p* < 0.05 was considered statistically significant.

**Table 5 jcm-11-06230-t005:** Characteristics of PAD in patients with CN and PAD.

Variable	Values (N and %)
Vessels affected	2.9 (min 1–max 6)
Femoral-popliteal axis	19/24 (79.2%)
Iliac artery	0/24 (0%)
Common femoral artery	0/24 (0%)
Superficial femoral artery	19/24 (79.2%)
Popliteal artery	8/24 (33.3%)
BTK arteries	18/24 (75%)
Anterior tibial artery	14/24 (58.3%)
Posterior tibial artery	12/24 (50%)
Peroneal artery	7/24 (29.2%)
BTA arteries	4/24 (16.7%)
Baseline TcPO2 (mmHg)	33 ± 5
Revascularization for healing ulceration (yes)	14/24 (58.3%)

BTK: below-the-knee; BTA: below-the-ankle.

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
