# Peer review of "The Neuro-Ischaemic Charcot Foot: Prevalence, Characteristics and Severity of Peripheral Arterial Disease in Acute Charcot Neuro-Arthropathy"

_jcm, 2022, doi:10.3390/jcm11216230_

Round 1

Reviewer 1 Report

The manuscript is interesting in regard to the topic addressed, with a not so small case history.

The metodological approach is correct and the conclusions are in line with the results obtained.

It could be worthy of publication, being very relevant to the important issue, however an extensive linguistic revision by a native English speaker or Editing system is a priority (there are many grammatical errors in the manuscript, the English used is not always scientifically correct).

If useful I recommend the following citations: DOI: 10.4081/or.2020.8670

Author Response

The manuscript is interesting in regard to the topic addressed, with a not so small case history.

 The metodological approach is correct and the conclusions are in line with the results obtained.

 It could be worthy of publication, being very relevant to the important issue, however an extensive linguistic revision by a native English speaker or Editing system is a priority (there are many grammatical errors in the manuscript, the English used is not always scientifically correct).

 If useful I recommend the following citations: DOI: 10.4081/or.2020.8670

Dear Reviewer

Thank you for your comments and appreciation. The text has been revised again by a native English speaker, precisely Dr. Raju Ahluwalia which is the 2nd Author.

Reviewer 2 Report

1.     The study The Neuro-Ischaemic Charcot Foot: prevalence, characteristics and severity of peripheral arterial disease in Acute Charcot Neuro-arthropathy” evaluate the longitudinal outcomes of patients affected by Charcot neuro-arthropathy (CN) with peripheral arterial disease (PAD) assessed the impact of PAD on CN patients on disease severity. 

2.     The present retrospective study involves two key groups, i.e. Charcot neuro-arthropathy (CN) and peripheral arterial disease (PAD). The study intended to measure the interaction between two groups on the severity of a disease outcome. Appropriate use of a statistical tool is the major driver in such analysis. However, the study suffers in data representation and analysis.

3.     The major problem in this manuscript is with statistical data analysis. Each table applied different statistics and thus must be mentioned. Nowhere is it mentioned what is the relation of p-values with experimental groups.

4.     There is no statistical correlation presented with variables. Neither experimental group were compared and analysed, e.g. two-way ANOVA for evaluating the interaction between groups

5.     Table 1; what is the meaning of the whole population? It seems to refer to the selected patients.

6.     CN with or without PAD, i.e. CN-PAD and CN+PAD were not compared; Again, diabetes prevalence in CN type1 and type 0 ranged from 80.9 to 89% (Table 2), indicating diabetes is the major factor in these patients. No correlation with the variable (diabetes) that connects CN and PAD.

7.     Diabetes is the primary cause of the pathogenesis of PAD. The Hb1c levels of these subjects (76) could have been a significant stratification of how these patients managed their glucose levels since 86.6% are diabetic. The homogeneous inclusion criteria and appropriate diagnostic methods are essential to detect PAD.

8.     Peripheral artery disease (PAD) is the reduced blood flow in the limbs due to narrowed arteries. Interestingly, neuro-ischaemic CN showed arterial occlusions of vessels predominantly below-the-knee arteries (75%) but not below-the-ankle arteries, indicating the role of limb muscle insulin resistance. The possible mechanism that connects CN with PAD needs to be explained thoroughly.

9.     A layout of the study design, including exclusion and inclusion criteria, could simplify to follow the work.

10.  There are no reasons to integrate Table 1 with Table 1a; 

11.  Table 4- the reasons for having such a Table are not understood. It is defined evaluation of samples in terms of absolute values and percentages. 

12.  Line 27- end-stage-renal-disease –case vs control value missing.

13.  The abstract needs to add one statement on the statistical approach used in the study. Since the sample size is small, it is unknown if the sampling followed a normal distribution.

Author Response

Dear Reviewer

Many thanks for your kind revision and suggestions.

Hereafter, authors reply point by point.

  1. The study “The Neuro-Ischaemic Charcot Foot: prevalence, characteristicsand severity of peripheral arterial disease in Acute Charcot Neuro-arthropathy” evaluate the longitudinal outcomes of patients affected by Charcot neuro-arthropathy (CN) with peripheral arterial disease (PAD) assessed the impact of PAD on CN patients on disease severity. 

Authors reply: no reply required

  1. The present retrospective study involves two key groups, i.e. Charcot neuro-arthropathy (CN) and peripheral arterial disease (PAD). The study intended to measure the interaction between two groups on the severity of a disease outcome. Appropriate use of a statistical tool is the major driver in such analysis.However, the study suffers in data representation and analysis.

Authors reply: Thank you for this comment which allows also to better clarify in the other revision points. The study aims to evaluated two groups: patients affected by Charcot Neuro-Arthropathy (CN) with or without Peripheral Arterial disease (PAD). Baseline characteristics and outcomes of interest were compared between groups.

The section related has been revised as reported below to be clearer for readers:

“Statistical analysis was performed by SAS (JMP12; SAS Institute, Cary, NC) for the personal computer. The sample size was calculated by the power analysis by adopting the two-tailed test of the null hypothesis with alfa= 0.05 and a value of beta= 0.10 as the second type error and, therefore, a test power equal to 90%. Continuous variables were expressed as the mean ±SEM. The Shapiro-Wilk test was used to statistically assess the normal distribution of the data. Comparisons between continuous variables were performed using the independent Student t-test. Categorical data were analyzed using the chi square test. P<0.05 was considered statistically significant.”

  1. The major problem in this manuscript is with statistical data analysis. Each table applied different statistics and thus must be mentioned. Nowhere is it mentioned what is the relation of p-values with experimental groups.

Authors reply: As reported above in point 2, the section on the statistical analysis has been clarified. The same method of analysis has been used in each table. As suggested, the relation of p value (<0.05) was reported in each table.  

  1. There is no statistical correlation presented with variables. Neither experimental group were compared and analysed, e.g. two-way ANOVA for evaluating the interaction between groups.

Authors reply: as reported in the text, the two main groups (CN with PAD and CF without PAD) were compared in terms of demographic and clinical baseline characteristics at the assessment and in terms of outcomes. As expected, there were some significant statistically difference at the baseline regarding some variables including cardiovascular risk factors, specifically patients with CN and PAD reported higher rate of cardiovascular risk factors than CN without PAD (see results section and table 2-4)).

  1. Table 1; what is the meaning of the whole population? It seems to refer to the selected patients.

Authors reply: Yes, the whole population is referred to patients included in the study. To be clearer, in each table, the word “included” was added to the “whole population” definition. In addition, under the text in the specific column, the number of patients regarding the whole population and the two study groups has been reported. Thank you for this suggestion and avoid confusion.

  1. CN with or without PAD, i.e. CN-PAD and CN+PAD were not compared; Again, diabetes prevalence in CN type1 and type 0 ranged from 80.9 to 89% (Table 2), indicating diabetes is the major factor in these patients. No correlation with the variable (diabetes) that connects CN and PAD.

Authors reply: Dear reviewer, in the current study all patients included were affected by CN; the aim was to evaluate the prevalence of PAD in this specific population and its impact on outcomes. Patients with CN and PAD were compared to those with CN without PAD aiming to evaluate the differences between groups in clinical practice and the potential prognosis of both groups (to directly highlight the potential impact of PAD).

For that concerning the other points, all patients included were affected by diabetes. To avoid any kind of confusion for readers, this point has been now clarified in Methods section, “patients selection” sub-section as follows: “This study is a retrospective study conducted in a single tertiary level DFS, assessing consecutive patients since January 2019 to February 2022, diagnosed with diabetes and acute CN stage 0 and 1 were included.  Patients excluded were patients without diabetes, those presenting with an extensive tissue loss and infection or those with fractures or dislocations of the osseous structures not suitable for limb salvage, and those with chronic CN”.

Regarding the diabetes prevalence in CN type 1 and type 0, the percentage you indicated (80.9 and 89%) were exclusively related to the diabetes type (type 2 vs type 1) as reported inside the brackets in the respective row. Anyway, this table has been removed as suggested.

  1. Diabetes is the primary cause of the pathogenesis of PAD. The Hb1c levels of these subjects (76) could have been a significant stratification of how these patients managed their glucose levels since 86.6% are diabetic. The homogeneous inclusion criteria and appropriate diagnostic methods are essential to detect PAD.

Author reply: As reported above in point 6, all patients included in the current study were affected by diabetes. As suggested, we have added the baseline Hb1c levels of the whole population, CN with PAD and CN without PAD groups.

  1. Peripheral artery disease (PAD) is the reduced blood flow in the limbs due to narrowed arteries. Interestingly, neuro-ischaemic CN showed arterial occlusions of vessels predominantly below-the-knee arteries (75%) but not below-the-ankle arteries, indicating the role of limb muscle insulin resistance. The possible mechanism that connects CN with PAD needs to be explained thoroughly.

Authors reply: in the Authors opinion, the explanation on PAD development in CN may be found not specifically in the CN per sé but in the concomitant cardiovascular risk factors. Below-the-ankle arterial disease appears to be present approximately in 40-50% of patients with diabetic foot ulcers and it is often related to the concomitant presence of End Stage Renal Disease. In the study group of the current study, only 3 patients were affected by ESRD and this data may reflect the rare presence of below-the-ankle arterial disease. Definitely, Authors retain the CN diabetic patients may develop PAD such as diabetic patients without CN according to the presence or not of cardiovascular risk factors, and PAD can’t be considered a protective factor for CN..

This point has been explained in the discussion section as follow:

Therefore, the authors retain that PAD may develop in patients with diabetes and CN such as in those with diabetes without CN usually in relation to the presence of concomitant cardiovascular risk factors such as hypertension, dyslipidemia, renal failure, smoke. Finally, the less severity of PAD we reported in our sample, specifically documented by the few cases of BTA arterial disease, may be related to the very low rate of ESRD patients. The involvement of BTA arteries in patients with diabetes is usually related to the concomitant presence of ESRD as reported in previous study [10,11,30].”

  1. A layout of the study design, including exclusion and inclusion criteria, could simplify to follow the work.

Author reply: inclusion criteria and exclusion criteria are better reported in the text (methods section) as follow “This study is a retrospective study conducted in a single tertiary level DFS, assessing consecutive patients since January 2019 to February 2022, diagnosed with diabetes and acute CN stage 0 and 1 were included.  Patients excluded were patients without diabetes, those presenting with an extensive tissue loss and infection or those with fractures or dislocations of the osseous structures not suitable for limb salvage, and those with chronic CN.”

A layout has been added as suggested (see Table 1).

  1. There are no reasons to integrate Table 1 with Table 1a; 

Author reply: table 1 has been removed as suggested, and the rate of Charcot type 0 and 1 included in the previous Table 1, now named Table 2.

  1. Table 4- the reasons for having such a Table are not understood. It is defined evaluation of samples in terms of absolute values and percentages. 

Author reply: the Author added this table and related data because it may easily to identify the characteristics of patients with CN and PAD we included in the study. To know the characteristics of PAD may be also useful to compare the data of the current study with data reported in literature, particularly the distribution arterial lesions and its severity.

  1. Line 27- end-stage-renal-disease –case vs control value missing.

Author reply: the percentage values of control is 0, the symbol “%” was missing and now added.

  1. The abstract needs to add one statement on the statistical approach used in the study. Since the sample size is small, it is unknown if the sampling followed a normal distribution.

Authors reply: A statement regarding the statistical approach has been added in the abstract.

Round 2

Reviewer 2 Report

The revised version of the manuscript has improved with better clarity.

The major aim is to map the prevalence and characteristics of the disease, which is important considering its management over the course of time.

Table 2-4 Here prevalence is expressed out of total samples and their percentage is included in parenthesis.
It is not clearly written with whom p<0.05 was considered statistically significant.

Line 258 Remove "smoke" as this is an external factor

Author Response

Dear Reviewer, many thanks for your comments.

The revised version of the manuscript has improved with better clarity.

Authors reply: thank you

The major aim is to map the prevalence and characteristics of the disease, which is important considering its management over the course of time.

Table 2-4 Here prevalence is expressed out of total samples and their percentage is included in parenthesis.
It is not clearly written with whom p<0.05 was considered statistically significant.

Authors reply

The prevalence of each column, expressed in absolute value and in percentage, is related to number of patients included (as reported as the top of each column) respectively in the whole population, PAD group and not PAD group.

In table 2-4, it has been now reported whom p<0.05 was considered statistically significant. The tables caption reported as follows "In the comparison between groups (CN with PAD vs CN without PAD) p<0.05 was considered statistically significant."

Line 258 Remove "smoke" as this is an external factor

Authors reply: "smoke" has been removed as suggested.